# Self-Reported Baseline Quality of Life Mirrors Treatment-Specific Characteristics of Cancer Patients

**DOI:** 10.3390/cancers15245763

**Published:** 2023-12-08

**Authors:** Anja Thronicke, Shiao Li Oei, Gerrit Grieb, Patricia Grabowski, Juliane Roos, Friedemann Schad

**Affiliations:** 1Research Institute Havelhöhe gGmbH, Hospital Gemeinschaftskrankenhaus Havelhöhe, 14089 Berlin, Germany; anja.thronicke@havelhoehe.de (A.T.);; 2Department of Plastic Surgery and Hand Surgery, Hospital Gemeinschaftskrankenhaus Havelhöhe, 14089 Berlin, Germany; 3Interdisciplinary Oncology and Palliative Care, Hospital Gemeinschaftskrankenhaus Havelhöhe, 14089 Berlin, Germany

**Keywords:** self-reported quality of life, prospective study, oncological patients

## Abstract

**Simple Summary:**

Contradictory research exists on the association of self-reported quality of life at tumor diagnosis (baseline quality of life, bQL) with treatment and disease-specific characteristics in oncological patients. The aim of our prospective study was to examine the role of bQL as a treatment predictor in oncological patients. We show that self-reported pain and low physical functioning or financial burden at tumor diagnosis were linked to lower systemic treatment, reduced surgery or reduced oncological treatment compliance, respectively. Thus, we found for the first time that survival-relevant quality-of-life variables reported during diagnosis are significantly associated with the subsequent oncological treatment of oncological patients. In addition, our results indicate that these treatment patterns were distinguishable between genders, as well as younger or older ages. Linking survival-predicting baseline QL scores to treatment-specific patterns emphasizes the importance of patient baseline physio-emotional state over the course of the disease and paves the way for early integration of patient-reported outcomes into oncological supportive concepts.

**Abstract:**

Background: Baseline quality of life (bQL) has been shown to be a predictor of the clinical outcome of oncological patients. The primary objective of the present study was to examine the role of bQL as a treatment predictor in oncological patients. Methods: In this prospective study, all-stage cancer patients registered in the Network Oncology registry were enrolled, and their bQL at diagnosis was evaluated. Results: Five hundred and thirty-eight oncological patients were eligible (median age 64 years). We show that survival-predicting bQL variables such as pain, low physical functioning or financial burden at tumor diagnosis were linked to lower systemic treatment (*p* = 0.03), reduced surgery (*p* = 0.007) or reduced oncological treatment compliance (0.01), respectively. Lastly, female gender and older cancer patients exhibited a tempered bQL. Conclusion: Our study is one of the first to reveal that bQL at tumor diagnosis is significantly associated with the prediction of oncological treatment with distinctive age- and gender-related patterns. Our results emphasize the need to address the physical, psychosocial, and financial burden of cancer patients prior to their oncological treatment with respect to age and gender. The associations found here pave the way for early integration of patient-reported outcomes into oncological supportive concepts.

## 1. Introduction

Patient-reported outcome measures, including self-reported baseline quality of life (bQL), play a decisive role in the context of early and comprehensive support for cancer patients as suggested by national guidelines [1,2]. For more than a decade, it has been known that bQL—the physical, social and emotional state of a cancer patient during the diagnosis—may forecast mortality and seems to be one of the most reliable predictors of survival [3,4,5,6,7,8]. An underlying rationale may be that patients with a better initial state of general symptoms and psychosocial well-being at diagnosis have advantages in keeping this state, comply better with standard oncological treatment and survive longer. In addition, bQL has been linked to the determination of post-operative complications across various cancer types [9,10]. However, little systematic research has been carried out so far on the association of bQL with treatment-specific characteristics, and there are conflicting opinions insofar as the literature reveals that bQL seems to be predictive [5] and, on the other hand, seems not to be associated with planned treatments [11]. Thus, the objective of the present prospective real-world data study was to examine the role of bQL in the subsequent treatment in all-stage oncological patients. We concentrated not only on treatment patterns but on disease as a secondary goal, as well as age- and gender-specific patterns.

To supply high-quality standards and to make the results internationally comparable, the European Organization for Research and Treatment of Cancer Core Quality of Life (EORTC QLQ-C30) questionnaires were utilized in the present study [12]. Recent publications indicate that the index scores and the summary score of the QLQ-C30 and other questionnaires, such as the QLQ-HCC19, are prognostic [6,7,8].

## 2. Materials and Methods

### 2.1. Study Design, Patients and Primary Objective

A prospective, non-randomized, multicenter real-world study was conducted, and data from an oncological registry, Network Oncology, were analyzed. Patients who were 18 years or older, of both genders, who gave written consent, had a tumor diagnosis and their data documented in the Network Oncology were included in the analysis. Patients were excluded when written consent was not given, when self-reported health-related quality of life at diagnosis was not documented, or when demographic and treatment-related data were missing in the Network Oncology registry. The primary objective of the present study was the assessment of the role of bQL in the subsequent treatment in all-stage oncological patients.

### 2.2. Ethical Approval and Consent to Participate

The study has been approved by the ethics committee of the Medical Association Berlin (Berlin—Ethik-Kommission der Ärztekammer Berlin). The reference number is Eth-27/10. Written informed consent was obtained from all patients prior to study enrolment. The study complies with the principles of the Declaration of Helsinki.

### 2.3. Data Collection

Structured queries from patient records were performed using the Network Oncology (NO) registry for oncological patients with the International Classification of Diseases codes C15 (esophagus cancer), C16 (stomach cancer), C17 (small intestine cancer), C18 (colon cancer), C20 (rectum carcinoma), C24 (extrahepatic biliary duct), C25 (pancreatic cancer), C30 (nasal cavity and middle ear), C34 (lung cancer), C45 (mesotheliom) and C50 (breast cancer). Tumor stage at first diagnosis was defined as the earliest recorded stage within a month of the diagnosis date and was coded according to the Union for International Cancer Control (UICC) stages of the 8th edition of TNM Classification of Malignant Tumors. Demographic hospital-related data, such as diagnosis, demographic and medical data, as well as integrative oncological treatment, were retrieved from the Network Oncology registry. Self-reported health-related quality of life was assessed at tumor diagnosis utilizing the European Organization for Research and Treatment of Cancer Questionnaire C30 (EORTC-QLQ-C30). Scoring was performed in accordance with the EORTC-QLQ-C30 Scoring Manual.

### 2.4. Sample Size Determination

For a two-sided sample size test assuming a power of 90% and a level of significance of 5% as well as a small size effect (d = 0.2), a total of 264 patients would be needed to confirm a statistically significant treatment effect according to Schoenfeld et al. [13].

### 2.5. Statistical Analyses

All statistical analyses were conducted using the software R, version 4.1.3 (10 March 2022). Data are presented using descriptive statistics and normally distributed continuous data as the mean and standard deviation (SD) or 95% CI and skewed distributions by the median and 95% confidence interval (CI). Binary and categorical variables were presented as absolute and relative frequencies using counts and percentages. For the comparison of continuous variables between groups at baseline, the unpaired Student’s *t*-test for independent samples was used. Chi-square analyses were performed for the comparison of categorical baseline variables. Adjusted multivariable linear and logistic regression analyses were performed to analyze association factors. All tests were performed two-sided. *p*-values < 0.05 were considered as significant.

## 3. Results

In total, 538 eligible patients were included in the subsequent outcome analysis. Demographic and clinical characteristics of patients are shown in Table 1. The median age of the patients was 64 years, and 70.6% were female. The majority of patients were diagnosed with breast cancer (43%), followed by lung (17%), colon (15%), rectum (9%) and other cancer (15%; see Table 1). Other cancer includes esophagus cancer, stomach cancer, small intestine cancer, extrahepatic biliary duct, pancreatic cancer, nasal cavity and middle ear cancer and mesothelioma. Tumor UICC stage I and II were predominant and comprised more than 50%; see Table 1.

### 3.1. Oncological Treatment

The majority of the patients underwent surgery (81%); see Figure 1. Furthermore, 65% of the patients received systemic therapy, including chemotherapy or targeted therapy, and 44% underwent radiation. Thirty-four percent of all oncological patients underwent *Viscum album* L. (VA, European white berry mistletoe) therapy in addition to standard oncological treatment; see Figure 1.

The median number of different applied non-pharmacological interventions (NPIs) was six (IQR 1–8) with a maximum of 13 interventions; see Figure 2. The majority of enrolled patients underwent psycho-oncological therapy (67%), embrocations (57%), movement therapies such as dance (56%) or physiotherapy (54%), massages (49%) and music therapy (46%); see Figure 2. Drawing therapy (25%) and biographical review (25%) were utilized to a lesser extent. Only a small proportion (5%) of patients underwent breathing therapy.

### 3.2. Association between bQL and Standard-Oncological Treatment

A better baseline physical functioning (OR 1.025, 95% CI: 1.007−1.041, *p* = 0.007) was significantly associated with surgical intervention, while patients with baseline diarrhea (OR 0.9895, 95% CI: 0.9803−0.9988, *p* = 0.028) or baseline financial burden were significantly less likely to have received surgery (OR 0.991, 95% CI: 0.983−0.999, *p* = 0.03); see Table 2. Patients with baseline dyspnea received radiation significantly more often (OR: 1.008, 95% CI: 1.0005−1.0161, *p* = 0.04); see Table 2. Patients with baseline pain received significantly more (OR 1.009, 95% CI: 1.0008−1.0173, *p* = 0.03) and patients with a baseline financial burden less (OR 0.993, 95% CI: 0.987−0.999, *p* = 0.01) systemic therapy; see Table 2. No significant associations were observed for baseline emotional functioning with surgery (*p* = 0.634), radiation (*p* = 0.347) or systemic therapy (*p* = 0.187).

### 3.3. Association between bQL and Add-On Complementary Therapies

Adjusted multivariable regression analyses revealed that patients with good baseline global health (OR 1.025, 95% CI: 1.006−1.0446, *p* = 0.011), appetite loss (OR 1.012, 95% CI: 1.002−1.0221, *p* = 0.018) or good social functioning (OR 1.35, 95% CI: 1.024−1.775, *p* = 0.034) were significantly more likely to undergo NPIs. On the contrary, patients with baseline insomnia underwent NPIs less often (OR 0.99, 95% CI: 0.982−0.998, *p* = 0.015); see Table 2.

Patients with good baseline social functioning (OR 1.39, 95% CI: 1.104−1.753, *p* = 0.005) received embrocations significantly more often; see Table 2. Multivariable regression analyses adjusting for age, gender, tumor type and bQL revealed that patients with good baseline social functioning (OR 1.4, 95% CI: 1.1−1.7, *p* = 0.004) had significantly higher odds of undergoing physiotherapy; see Table 2. Patients with a good baseline social functioning (OR: 1.24, 95% CI: 1.002−1.543, *p* = 0.048) received significantly more add-on VA therapy; see Table 2. Patients with baseline fatigue (OR 1.016, 95% CI: 1.04−1.029, *p* = 0.01) and patients with a good baseline social functioning (OR 1.24, 95% CI: 1.002−1.5339, *p* = 0.047) received massages significantly more often; see Table 2. Cancer patients with baseline fatigue (OR 1.03, 95% CI: 1.015−1.045, *p* = 0.0001) and patients with good baseline social functioning (OR 1.46, 95% CI: 1.14−1.97, *p* = 0.003) underwent a biographical review significantly more often; see Table 2. In addition, patients with baseline fatigue sought psychological care significantly more often (OR 1.01, 95% CI: 1.0−1.02, *p* = 0.04); see Table 2. Patients with baseline dyspnea underwent drawing therapy significantly less often (OR 0.991, 95% CI: 0.983−0.999, *p* = 0.028; see Table 2), while the probability for female patients was almost two times higher (OR 1.9, 95% CI: 1.0−3.5, *p* = 0.03; see Figure 3). In addition, better baseline social functioning (OR 1.413, 95% CI: 1.112−1.796, *p* = 0.0047) and higher baseline financial burden (OR 1.007, 95% CI: 1.0006−1.0132, *p* = 0.032) were significantly associated with participation in drawing therapy; see Table 2. No significant associations were observed for baseline emotional functioning with any of the analyzed complementary, integrative therapies, including *Viscum album* L.

### 3.4. Gender- and Age-Mediated Association Factors

Female patients (OR 2.2, 95% CI: 1.3−3.7, *p* = 0.002), patients with fatigue (OR 1.01, 95% CI: 1.0−1.0, *p* = 0.02), and patients with better baseline social functioning (OR 1.3, 95% CI: 1.1−1.7, *p* = 0.01) had significantly higher odds of taking part in a dance therapy; see Table 2 and Figure 3. Music therapy was significantly more often (three times higher probability) used by female patients (OR 2.5, 95% CI: 1.4−4.3, *p* = 0.001) compared to male patients; see Figure 3. Furthermore, female patients were almost two times more likely to undergo drawing therapy (OR 1.9, 95% CI: 1.0−3.5, *p* = 0.03); see Figure 3. Female patients with a significant two times higher probability (OR 2.2, 95% CI: 1.3−3.6, *p* = 0.002) received massages more often; see Figure 4. Female patients (two times higher probability) underwent ECLR/biographical review (OR 2.3, 95% CI: 1.3−4.2, *p* = 0.006) significantly more often and sought psycho-oncological care (OR 2.0, 95% CI: 1.2−3.2, *p* = 0.008) more often than male patients; see Figure 3.

Younger oncological patients revealed a significant association with a higher financial burden (ß = −0.686, *p* = 5.67 × 10^−7^), i.e., per year of lower age baseline financial burden would increase by 68.6%; see Figure 4. There was also a positive, highly significant association between older age and improvement of baseline social functioning, i.e., per year of age, social functioning would increase by 53% (ß = 0.53, *p* = 9.53 × 10^−5^); see Figure 4.

Finally, and importantly, older age of oncological patients was significantly associated with lower baseline fatigue (ß = −0.24, *p* = 0.03), i.e., each year of aging improves baseline fatigue by 24%; see Figure 4. Younger age was significantly associated with the application of add-on *Viscum album* L. therapy (ß = −0.004, *p* = 0.029), i.e., each year of age decreases the application by 0.4%; see Figure 4.

## 4. Discussion

The findings of the present study reveal association patterns between bQL and respective treatment patterns of tumor patients. This is in line with another publication where the authors acknowledge that bQL data seem to provide the most reliable information to establish prognostic criteria for the treatment of patients with cancer as their assessment seems to be easier compared to follow-up QL evaluations [5]. On the other hand, other authors state that no association was found between patient bQL and planned treatment [12]. In our study, we determined that cancer patients with good physical functioning received significantly more surgical procedures. As baseline physical functioning was shown to predict good overall survival [8], and since surgery plays an important role in the improvement of overall survival in many cancer patients [14,15], one can possibly conclude that good physical functioning, surgery and improved survival may be interconnected. This perhaps needs to be kept in mind with patients where surgery is precluded or where physical functioning is low at the time of diagnosis, as these patients need to be more closely monitored due to possibly deteriorated survival outcomes. The findings of our study also revealed that cancer patients with dyspnea significantly more often received radiation than non-dyspneic patients. Radiation is performed in early-stage lung cancer as a curative approach, especially in patients who are not expected to receive surgery or to reduce the tumor burden and improve QL, e.g., in patients with more advanced lung cancer as a means of palliative care [16]. If initiated promptly in advanced lung cancer, radiation can improve the QL of patients [16]. Thus, baseline dyspnea and radiation are another association pattern found in our study and may be associated with an improved QL. Tumor-related pain represents a prominent and survival-associated cancer symptom and was shown in our study to be associated with an increased application of systemic therapy. Since the increased training and education of health professionals could improve cancer pain management [17], further research is needed to unravel the relationship between baseline pain, systematic treatment and survival outcomes.

Another interesting baseline QL variable in our study was self-reporting of diarrheic symptoms at diagnosis, which was shown to be negatively associated with surgery. A population registry study found that diarrhea in cancer patients was one of the symptoms with the highest impact on QL and revealed a key management issue in palliative and cancer care [18]. The prevalence of diarrhea in palliative care is about 20% [19], and the ESMO clinical practice guideline suggests that patients should be asked for their history of diarrhea, incontinence and gastrointestinal diseases during assessment as they may be reluctant to report it voluntarily [20]. As surgery is associated with improved overall survival in many cancers, it is mandatory to further study the association between the occurrence of baseline diarrheic symptoms, reduced surgery options and overall survival.

Moreover, we found that patients with baseline financial burden received less surgery and less systemic therapy independently of age or gender. In line with our observations, the results of a systematic review revealed that financial burden was linked to nearly twice the odds of non-adherence with cancer treatment [21] and to major morbidity [22]. Financial burden has gained importance during the last decade as it has been detected to also disproportionately affect younger and socioeconomically disadvantaged cancer patients and has been linked to impaired QL [21]. In a previous study, the findings of our research group revealed that financial burden correlated with baseline pain, anxiety, low mood and younger age in early and late-stage lung cancer [23]. Thus, as the financial burden in cancer patients correlates significantly with less applied standard oncological therapies, it is suggested that these patients, especially younger patients or those with emotional instabilities, need to be screened and monitored closely. It may be assumed that the financial burden at diagnosis acts as a surrogate marker for the emotional state of oncological patients. Answering questions about their financial burden may circumvent direct answers on their self-reported emotional state [23] or their need for psychological care. This could comply with findings that a number of individuals indicate that they do not need psychological care [24] or feel ashamed or guilty when diagnosed with cancer and may be reluctant to seek external psychological help [25]. Another explanation could be adaptive mechanisms in cancer patients [26]. Both findings may support the observations in our study that emotional functioning at diagnosis was not linked to any subsequent standard or complementary oncological treatments, including psychotherapy. However, as about one-third of oncological patients experience distress [27] or financial burden [23], special focus should be placed on screening for patients under financial and/or emotional stress.

We found that bQL, besides being a predictor of standard oncological treatment, also predicted the use of non-pharmacological interventions in cancer patients. For instance, good baseline social functioning predicted the application of complementary therapy, such as mistletoe therapy, and non-pharmacological interventions, such as physiotherapy, biographical review or embrocations, a care application that is especially suitable for bedridden people. Good baseline social functioning was also associated with art therapies such as drawing or dance therapy and further with massage applications. In our study, we show that cancer patients constantly and to a great extent participate in non-pharmacological integrative oncology concepts, including art therapies, massages and psycho-oncological as well as biographical therapies. Integrative oncology has been established and acknowledged for the improvement of QL during recent decades in international academic and public cancer centers [28,29,30]. The demand for integrative oncology, comprising complementary therapies in addition to standard oncological care, is on the rise due to the increasing self-awareness of people with cancer and cancer survivors who want to self-manage their own needs as active participants in the treatment [31,32]. Thus, the intention is that these concepts are accepted in the future as routinely applied healthcare elements as they contribute to clinical effectiveness by helping people with cancer maintain their QL as well as dosing and procurement of their standard oncological treatment.

Lastly, our study showed that the oncological treatment pattern was not only determined by the bQL but also by patient age and gender. Growing age was shown in our study to be linked to a significantly better baseline fatigue, significantly lower financial burden and a significantly better baseline social functioning. On the other hand, older patients underwent an add-on *Viscum album* L. therapy to a lesser extent, in line with other publications that younger oncological patients undergo more complementary therapies [33]. Female patients were shown to be significantly more likely to take part in art therapies, including music, drawing or dancing therapies and received significantly more massage therapies than male cancer patients. It is important to note that psycho-oncological care or biographical reviews were also sought at a significantly higher rate by female cancer patients. Data from a meta-analysis revealed that the female gender seems to be another positive predictor of survival [8]. Consistent with this, our data suggest that female cancer patients are much more likely to actively seek or use art therapies, body therapies or psycho-oncological therapies compared to their male counterparts, which may lead to a better QL and a better compliance with standard oncology treatment.

Our study is limited by the non-randomized nature of our analysis, which is prone to unwanted selection biases; however, this was counteracted by the application of adjusted multivariable regression analysis. Furthermore, other selection biases could have been introduced as it could be anticipated that healthier patients or patients undergoing integrative oncological therapies are more open to self-reporting in quality-of-life questionnaires. Another limitation is the observational character of the present study. Thus, our findings and conclusions must be handled with care. However, the study presented here reveals the real-world situation in oncological health services research and complements the controversial discussion and the importance of baseline quality of life representing a predictor of oncological therapies.

## 5. Conclusions

In summary, our study is one of the first prospective studies to assess the prognostic value of bQL in oncological patients for treatment patterns. We show that baseline scores, such as diarrhea and the survival predictor of low physical functioning, were linked in our study to lower surgery rates, while higher baseline financial burden correlated with reduced oncological treatment compliance. In addition, we found that treatment patterns also distinguished between gender and age. We suggest monitoring younger cancer patients due to a deteriorated bQL pattern at diagnosis. Furthermore, patients with survival-predicting bQL scores, such as low physical functioning, pain or appetite loss, need to be closely monitored at the time of diagnosis. Financial burden, which is especially known to disproportionately affect younger and socioeconomically disadvantaged cancer patients, needs to be examined at diagnosis since it is linked to major morbidity and has been shown in our study to be associated with non-compliance with surgery and systemic therapy. Linking survival-predicting baseline QL scores to treatment-specific patterns emphasizes the importance of patient baseline physio-emotional state over the course of the disease. As accumulating data in the literature indicate an association between bQL and clinical outcomes, such as overall survival, our results emphasize the need to examine and address physical and psychosocial needs as well as financial burdens before oncological treatment.

## Figures and Tables

**Figure 1 cancers-15-05763-f001:**
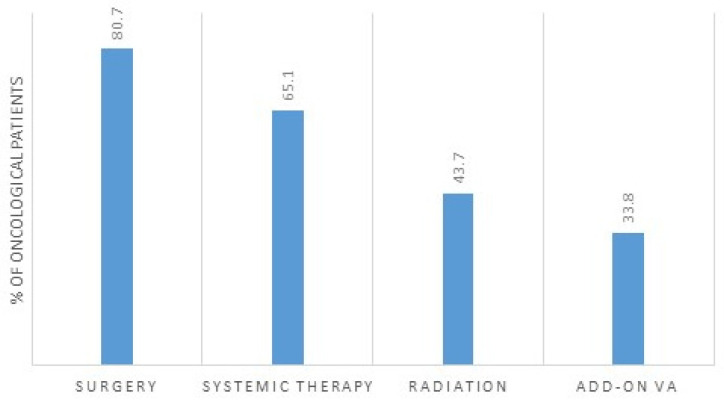
Oncological treatment and add-on treatment, *n* = 538 oncological patients. VA, *Viscum album* L.

**Figure 2 cancers-15-05763-f002:**
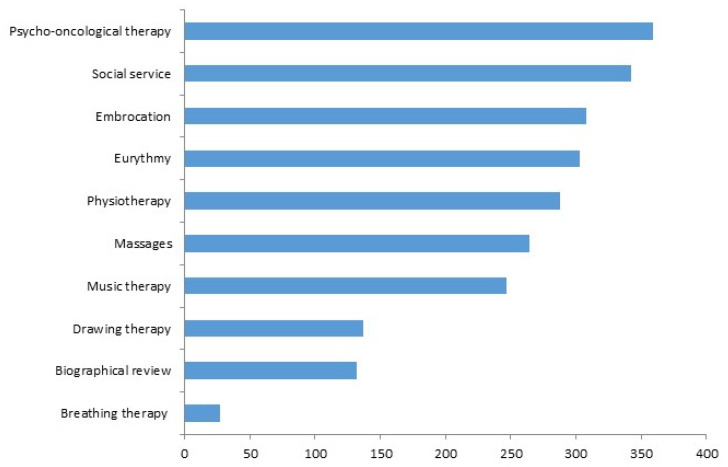
Non-pharmacological oncological treatment, *n* = 538 oncological patients.

**Figure 3 cancers-15-05763-f003:**
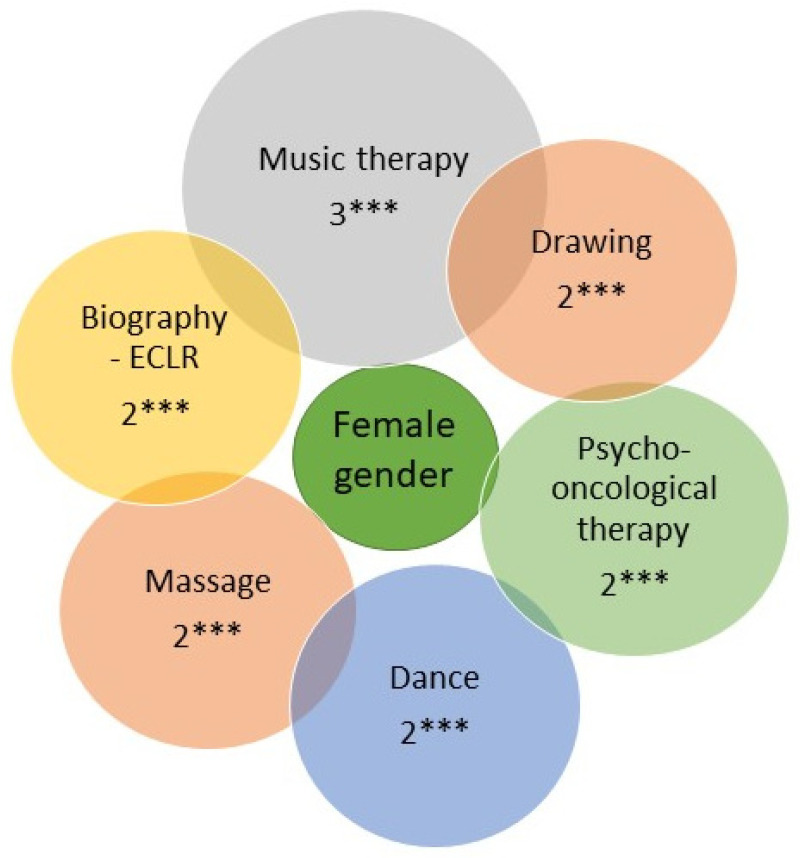
Female-gender-associated applications of non-pharmacological therapies. Values are indicated as estimates with *p*-value, reference: male gender; ECLR, elaborate consultation and life review. The size of the bubbles corresponds to the association or estimate size; ***, *p* < 0.005.

**Figure 4 cancers-15-05763-f004:**
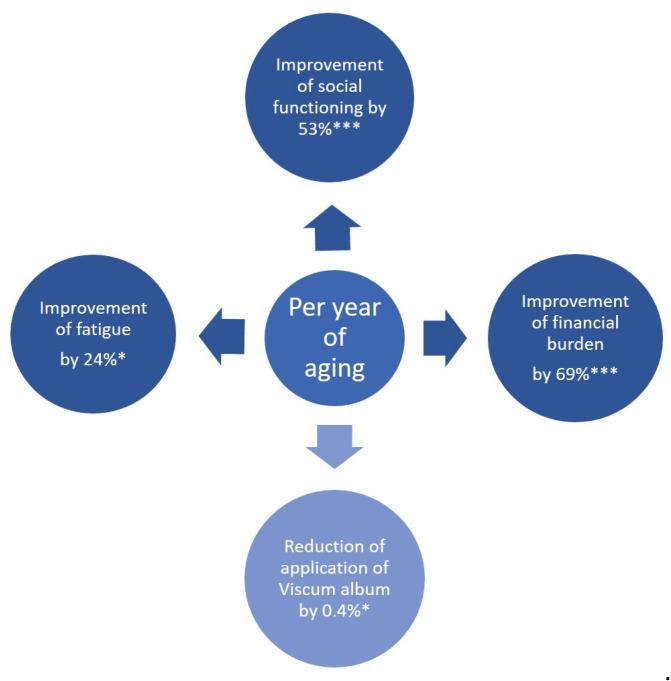
Age-associated bQL and treatment. *p*-value, *, *p* < 0.05; ***, *p* < 0.005; bQL, baseline quality of life (at time of diagnosis).

**Table 1 cancers-15-05763-t001:** Baseline characteristics of oncological patients with records of self-reported baseline quality of life.

	All Patients (*n* = 538)
Age in years, median (IQR)	65 (54.0−72.0)
Gender, female, *n* (%)	382 (71.0)
Gender, male, *n* (%)	156 (29.0)
UICC 0, *n* (%)	14 (2.6)
UICC I, *n* (%)	128 (23.8)
UICC II, *n* (%)	148 (27.5)
UICC III, *n* (%)	88 (16.4)
UICC IV, *n* (%)	70 (13.0)
Breast cancer	233(43.3)
Lung cancer	92 (17.1)
Colon cancer	82 (15.2)
Rectum cancer	49 (9.1)
Other cancer	82 (15.2)

Percentages of sub-characteristics may not add up to 100% due to rounding of the numbers; *n*, number of patients; %, percent; IQR, interquartile range. UICC, UICC and TNM classification of malignant tumors.

**Table 2 cancers-15-05763-t002:** bQL—associated application of standard oncological care and complementary therapy. Multivariable regression analyses adjusting for age, gender, tumor type and bQL. Values are indicated as odds ratios with *p*-values.

	SURG	RAD	SYS	VA	NPI	PHY	DAN	EMB	DRA	MAS	ECLR	PSY
Global health	--	--	--	--	1.03 *	--	--	--	--	--	--	--
Appetite loss	--	--	--	--	1.01 *	--	--	--	--	--	--	--
Insomnia	--	--	--	--	0.99 *	--	--	--	--	--	--	--
Fatigue	--	--	--	--	--	--	1.01 *	--	--	1.02 *	1.03 ***	1.01 *
Social functioning	--	--	--	1.24 *	1.35 *	1.4 ***	1.3 *	1.39 **	1.4 **	1.24 *	1.46 ***	--
Phys. functioning	1.03 **	--	--	--	--	--	--	--	--	--	--	--
Pain	--		1.01 *	--	--	--	--	--	--	--	--	--
Financial burden	0.99 *		0.99 *	--	--	--	--	--	1.01 *	--	--	--
Diarrhea	0.99 *		--	--	--	--	--	--	--	--	--	--
Dyspnea	--	1.01 *	--	--	--	--	--	--	0.99 *	--	--	--

*, *p* < 0.05; **, *p* < 0.01; ***, *p* < 0.005; --, no significant association detected; SURG, surgery; RAD, radiation; SYS, systemic therapy; VA, *Viscum album* L. therapy; NPI, non-pharmacological intervention; PHY, physiotherapy; DAN, dance; EMB, embrocation; DRA, drawing therapy; MAS, massage; PSY, psycho-oncological therapy; ECLR, biographical review.

## Data Availability

The datasets that support the findings in this article are not publicly available for privacy and security reasons but can be obtained from the corresponding authors upon reasonable request.

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
