# Peer review of "Self-Reported Baseline Quality of Life Mirrors Treatment-Specific Characteristics of Cancer Patients"

_cancers, 2023, doi:10.3390/cancers15245763_

Round 1

Reviewer 1 Report

Comments and Suggestions for Authors

Authors compiled a manuscript on topic “Self-reported baseline quality of life mirrors treatment-specific characteristics of cancer patients”. The study focusses on the assessment of base line quality of life of cancer patients and prediction of outcome of oncology treatment. Various parameters were included int the study to assess the patient’s outcome. Authors very well compiled the manuscript and can be published with addressing the following points:

1.     Please mention in the manuscript whether the data was collected only from patient records or patients were questioned and how?

2.     Calculation of sample size is not mentioned in the manuscript. Please add how you determined the sample size?

3.     Please add more details to the inclusion and exclusion criteria for patients.

4.     In the abstract and summary section patient psychology and emotions are considered as one of the factors to be examined but these are not listed in the result section. Psychology and emotional state are important parameters which may have their role in outcome of the treatment.

Author Response

Thank you for your careful review and your valuable comments. We have answered your query and have revised the manuscript according to your suggestions and comments. We hope that we could sufficiently answer all of your questions and remarks and hope that our changes will meet your approval.

Response to Reviewer 1 Comments

Reviewer 1: Authors compiled a manuscript on topic “Self-reported baseline quality of life mirrors treatment-specific characteristics of cancer patients”. The study focusses on the assessment of base line quality of life of cancer patients and prediction of outcome of oncology treatment. Various parameters were included in the study to assess the patient’s outcome. Authors very well compiled the manuscript and can be published with addressing the following points:

Point 1: Please mention in the manuscript whether the data was collected only from patient records or patients were questioned and how? Response 1: Thank you for the review of our manuscript. Patients were questioned and we indicated the procedure in the Material and Method section (page 2 and 3, line 95-98).

Point 2: Calculation of sample size is not mentioned in the manuscript. Please add how you determined the sample size? Response 2: We included a sample size calculation, see section ,methods, page 3, line 101-104.

Point 3: Please add more details to the inclusion and exclusion criteria for patients. Response 3: Thank you for your suggestions. The inclusion criteria were extended and the exclusion criteria were included, please see page 2, line 70-74.

Point 4: In the abstract and summary section patient psychology and emotions are considered as one of the factors to be examined but these are not listed in the result section. Psychology and emotional state are important parameters which may have their role in outcome of the treatment. Response 4: Thank you for this suggestion. We have now included in our results the information that we have not found for emotional functioning any association with a subsequent therapy, please see ,results, page 5, line 160-162 as well as page 9, line 198-200. However, for self-reported ,financial burden, which has been associated with anxiety and low mood in one of our previous studies (1), we observed associations with various subsequent treatment schemes in the present study. This self-reported parameter may seem to function as a kind of surrogate parameter for the emotional state of the oncological patients. We have now extended the discussion on this topic involving the emotional state of the patients, please see section ,discussion, page 8, line 318- 330. We hope you agree.

  • Thronicke A, von Trott P, Kröz M, Grah C, Matthes B, Schad F. Health-Related Quality of Life in Patients with Lung Cancer Applying Integrative Oncology Concepts in a Certified Cancer Centre. Evid Based Complement Alternat Med. 2020 May 10;2020:5917382. doi: 10.1155/2020/5917382. PMID: 32454866; PMCID: PMC7238336.

Reviewer 2 Report

Comments and Suggestions for Authors

Thank you very much for giving me the opportunity to review this manuscript. The work is well-written, very engaging to read, and highlights a fundamental aspect for the comprehensive care of oncology patients. In my opinion, the work can be accepted in its presented version.

Author Response

Dear reviewer,

Thank you very much for your review of our manuscript and your evaluation that it, is very well-written and very engaging to read,.

With kind regards,

The authors